# Detection Limits of Antibiotics in Wastewater by Real-Time UV–VIS Spectrometry at Different Optical Path Length

Feng Li [1], Xiaodong Wang [1,*], Manzi Yang [1], Ming Zhu [2], Wei Chen [2], Qiran Li [2], Delin Sun [3], Xuejun Bi [1], Zakhar Maletskyi [4] and Harsha Ratnaweera [4]

1   School of Environmental and Municipal Engineering, Qingdao University of Technology, Qingdao 266033, China
2   Shenzhen Institute of Advanced Technology, Chinese Academy of Sciences, Shenzhen 518055, China
3   Shandong Oubeier Software Technology Co., Ltd., Jinan 250021, China
4   Faculty of Science and Technology, Norwegian University of Life Sciences, P.O. Box 5003, 1432 Ås, Norway
*   Correspondence: wangxiaodong@qut.edu.cn

**Abstract:** Real-time monitoring of antibiotics in hospital and pharmaceutical wastewater using ultraviolet–visible (UV–Vis) spectroscopy is considered a promising method. Although gas chromatography–mass spectrometry (GC–MS) and other methods can detect antibiotics with quite low limits of detection (LOD), they possess various limitations. UV–Vis spectroscopy combined with chemometric methods is a promising choice for monitoring antibiotics. In this study, two immersed in situ UV–Vis sensors were used to explore the relationship between absorption spectra and antibiotics and study the influence of the optical path length on the LOD. The LODs of sensor 2 using a 10 cm optical path is up to 300 times lower than that of sensor 1 using a 0.5 mm optical path. Moreover, multiple antibiotics in the wastewater were investigated in real-time manner. The absorption spectra of 70 groups of wastewater samples containing different concentrations of tetracycline, ofloxacin, and chloramphenicol were measured. The results indicate that the nine wavelengths selected by interval partial least squares (iPLS) after the second derivative pretreatment have better predictability for ofloxacin and the six wavelengths selected by competitive adaptive reweighted sampling (CARS) after the first derivative. The multi-fold cross-validation results indicate that the model has a good predictive ability.

**Keywords:** UV–Vis spectrometry; partial least squares; antibiotics; online monitoring; hospital wastewater

## 1. Introduction

Since their inception, antibiotics have played an important role in treating of several common bacterial diseases. However, the irrational use of antibiotics is a major global public health concern. Hospitals and pharmaceutical plants produce large amounts of highly concentrated antibiotic wastewater daily [1]. Therefore, the online monitoring of antibiotics is a key development in the field of wastewater treatment [2]. Combining online monitoring instruments with traditional laboratory water quality detection methods can provide wastewater treatment managers with a convenient way to view real-time water quality indicators [3,4]. Many types of online monitoring instruments (e.g., pH, oxidation-reduction potential (ORP), ammonia nitrogen, and dissolved oxygen (DO) sensors) are expensive and require regular calibration and maintenance [5]. Recently, the combination of ultraviolet–visible (UV–Vis) spectrophotometry and chemometric models has become an efficient, low-cost, and highly sensitive method for in situ online monitoring of pollutants in wastewater [6]. The principle of UV–Vis spectroscopy is that when light passes through a solution, the degree of light absorption varies with different wavelengths [7]. This method is based on the Beer–Lambert law, which states that when monochromatic radiation passes through a solution of a substance to be measured, the amount absorbed by the substance within a certain concentration range is proportional to the concentration of the substance

and the thickness of the liquid layer (optical path length) [8]. Different types, numbers, and arrangements of functional groups in organic compounds lead to the formation of absorption peaks at different wavelengths. This is an effective method for analyzing the absorbance characteristics of antibiotics through experiments. The UV–Vis spectrum was then applied in this study to establish the online monitoring of antibiotics.

Methods for detecting antibiotics in wastewater can be approximately divided into the following two categories: traditional laboratory detection methods and in situ detection using monitoring equipment. The advantages of laboratory detection methods include high accuracy and low LOD. Yan et al. [9] used an ultra-performance liquid chromatography-tandem mass spectrometry (UPLC-MS/MS) system to detect the concentration of antibiotics at multiple sampling points in the Yangtze Estuary, and the LOD reached the order of ng/L. This method is particularly suitable for sampling in different seasons. Presently, the detection of organic compounds still relies on liquid chromatography (LC) or gas chromatography (GC) because these methods have good specificity and sensitivity. However, sample pretreatment such as solid-phase extraction (SPE) is usually required for these methods [10]. The disadvantages of this method include being expensive and not real-time [11]. Several studies have shown that it is feasible to use online spectroscopy, which is a sensitive, convenient, and efficient method, to monitor organic matter in water. [12]. With the advent of fixed optical array detectors, online spectroscopy has become feasible, and the combination of UV–Vis spectrophotometry and chemometric algorithms is regarded as a promising method. The principle of this method is that different functional groups of organic compounds generate absorption peaks at different wavelengths [13]. This type of UV–Vis spectrometry-based instrument is gradually being recognized and used due to its simple structure, fast real-time response, no secondary pollution, and low maintenance. However, a gap in LOD between spectrometry-based methods and GC- or LC-based methods remains, and saturated hydrocarbons and simple straight-chain alcohols are not absorbed in the UV–Vis range. Hence, the application scenario of spectrometry-based methods will be limited.

LOD is the lowest signal or corresponding quantity determined from the signal that can be observed with sufficient confidence or statistical significance [14]. Low LODs are required because organic matter concentrations are generally low in aquatic environments. Presently, the LOD for online monitoring of antibiotics in water by UV–Vis spectrophotometry is of the order of µg/L, which can meet the needs of wastewater with high organic concentrations such as hospital wastewater; however, this is unsuitable for rivers and lakes. Environmental quality standards for surface water (GB 3838-2002) [15] requires that the concentration of COD is not higher than that of 30 mg/L in quasi-IV water. Therefore, the removal of antibiotics with concentrations between 0.1–1 mg/L and 1–10 mg/L is crucial to meet sewage discharge standards [16].

Different water qualities require the suitable optical path length. This is because the optimum optical path length increases the stability and accuracy of the test [17]. An excessively long optical path will result in the array detector not receiving enough signal, while a short optical path will result in a higher LOD. Therefore, it is essential to choose the suitable optical path length according to water conditions and through experiments.

To improve the prediction accuracy, it is usually necessary to select effective bands in the ultraviolet and visible ranges for modeling [18]. Selecting wavelengths that are useful to the model can reduce the number of calculations and introduce errors into the model. The most commonly used wavelength selection algorithms select several bands, each of which still contains a large range of wavelengths. This leads to an overlap of similar wavelengths in the data, which is not conducive for simplifying the calculations or improving the accuracy of the model. Commonly used wavelength selection algorithms include the moving window partial least squares (MW-PLS) method, successive projection algorithm (SPA), uninformed variable elimination (UVE) algorithm, and competitive adaptive reweighted sampling (CARS) [19,20]. These are common methods for modeling the

absorption spectrum, including principal component analysis, partial least squares, and artificial neural networks.

In this study, the LODs of three typical antibiotics were determined by UV–Vis spectroscopy using two sensors, and a method combining UV–Vis spectroscopy with chemometric methods to monitor the concentration of antibiotics in wastewater samples was proposed. A model with better predictability was established by preprocessing the data and modeling after wavelength selection. The feasibility of applying UV–Vis spectroscopy to the online monitoring of antibiotics in hospital and pharmaceutical wastewater was investigated.

## 2. Materials and Methods

### 2.1. Subsection

The UV–Vis absorption spectra in this study were obtained using an in situ UV–Vis sensor (spectro::lyser V3, s::can Messtechnik, Vienna, Austria, collectively referred to as Sensor 1 in the following) and an underwater UV–Vis sensor (Jointly developed by Qingdao University of Technology and Chinese Academy of Sciences, collectively referred to as Sensor 2 in the following (see Figure A1 for the scheme)). The detailed data for the two sensors are shown in Table 1. For the Sensor 1, the absorbance values and spectra were presented on the screen through the moni::tool V3.1.4, and the data were exported to the computer through a mobile storage device to facilitate further analysis. Sensor 2 can be connected directly to a PC and controlled via LabVIEW 2018 software.

**Table 1.** Comparison of the parameters of the two sensors.

|  | Sensor 1 | Sensor 2 |
| --- | --- | --- |
| Spectrometer Probe |  |  |
| Lamp type | Xenon lamp | Xenon lamp |
| Optical path | 0.5 mm | 1–10 cm |
| Control platforms | Moni::tool V3.1.4 | LabVIEW 2018 |
| Resolution | 2.5 nm | 0.5 nm |
| Scan interval | 70 s | 30 s |
| Scan range | 200–737.5 nm | 186.5–665.5 nm |

### 2.2. Wastewater Samples

In this study, tetracycline, ofloxacin (quinolones), and chloramphenicol were selected and diluted to different concentrations in distilled water and wastewater. All antibiotics used were provided by Solarbio (Beijing Solarbio Science & Technology Co., Ltd., Beijing, China). The purities of tetracycline, ofloxacin, and chloramphenicol were 95, 99, and 99%, respectively. The absorption spectra of samples with different concentrations were measured using Sensors 1 and Sensors 2 (With four optical path lengths of 10 cm, 5 cm, 3 cm, and 1 cm). However, the two sensors differed in upper absorbance limits, and the ranges of sample concentration for their tests are also different. The range of sample concentration for Sensor 1 was set to 0–30 mg COD/L. Therefore, the corresponding concentration ranges of tetracycline, ofloxacin, and chloramphenicol are 0–16.03 mg/L, 0–14.27 mg/L, and 0–33.67 mg/L, respectively. For Sensor 2, the maximum absorbance in the near-UV region is around 1.5 when the optical path length is set to 10 cm. Thus, the concentration ranges for tetracycline, ofloxacin, and chloramphenicol are 0–1.6 mg/L, 0–3 mg/L, and 0–5 mg/L, respectively, based on the maximum absorbance. According to the same principle, the concentration ranges of the three antibiotics were 0–7.5 mg/L, 0–9 mg/L, and 0–10 mg/L when the optical path length is set to 5 cm; 15 mg/L, 20 mg/L, and 20 mg/L at 3 cm; and 25 mg/L, 30 mg/L, and 30 mg/L at 1 cm, respectively.

According to the Technical Guidelines for Hospital Wastewater Treatment of China [21], the COD, suspended solids (SS), and ammonia concentration range of hospital wastewater

is shown in Table 2. However, hospital wastewater is difficult to obtain owing to the public sanitation safety regulations. In this study, the antibiotics were diluted into the influent wastewater of wastewater treatment plants (WWTP) to simulate hospital wastewater. First, tetracycline, ofloxacin, and chloramphenicol solutions were prepared at a concentration of 200 mg/L, respectively, and then diluted 200 mg/L solutions with WWTP wastewater to make some of 0–25 mg/L solutions. These WWTP wastewater samples containing different concentrations of antibiotics were measured sequentially with Sensor 2. Guo et al. [22] investigated the average influent water quality of 127 wastewater treatment plants in China. In this study, the water quality of wastewater treatment plant and hospital wastewater was compared, and the results are shown in Table 2. Wastewater for this study was obtained from a wastewater treatment plant in Qingdao, Shandong Province, China, and the water quality was in accordance with the concentration range of the hospital wastewater. Specifically, the concentrations of COD, BOD, and ammonia nitrogen of the wastewater using in this study were in accordance with the concentration range of hospital wastewater, and only the concentration of SS was higher. The higher SS in the wastewater, the stronger the refraction and scattering effect on light. Therefore, the lower LOD will be obtained in hospital wastewater than in this study.

**Table 2.** Comparison of hospital wastewater, WWTP influent water, and the wastewater quality used in this study.

|  | COD (mg/L) | $BOD_5$ (mg/L) | SS (mg/L) | $NH_3–N$ (mg/L) |
|---|---|---|---|---|
| Hospital wastewater [1] | 150–300 | 80–150 | 40–120 | 10–50 |
| WWTP influent water [2] | 219.97 | 81.64 | 148.54 | 22.83 |
| Wastewater using in this study | 211 | 103 | 143 | 20.42 |

[1] The range of hospital wastewater quality indexes is from reference [21]. [2] The average of WWTP influent water quality indexes is from reference [22].

Ofloxacin, tetracycline, and chloramphenicol showed obvious spectral overlap. To explore the accuracy of antibiotic detection in wastewater samples, 70 groups of wastewater samples with three types of antibiotics at different concentrations were designed. The absorbance spectra of 70 wastewater samples were sequentially measured and a chemometric model was used to predict the concentrations of the three antibiotics.

*2.3. Modeling*

The three compounds had absorption peaks in the range of 250–310 nm, but the overlap of the absorption spectra was significant. It is impossible to determine the different components of a synthetic wastewater sample simultaneously using traditional spectrophotometry. Therefore, it was necessary to select an appropriate multivariate correction method. In addition, although the absorption peaks of tetracycline and chloramphenicol at 280 nm overlapped, they did not completely coincide. There are differences in the absorption peak positions and curve change trends of the three antibiotics, which indicates that the concentration of each component has a different contribution to the absorption spectrum of the mixed solution at different wavelengths. There were 81 wavelength variables in the wavelength range of 200–400 nm. Modeling with all wavelengths leads to problems such as model complexity or overfitting. Thus, to simplify the model and extract useful information from the spectrum, preprocessing and wavelength selection are required.

2.3.1. Data Preprocessing

The absorption spectrum obtained from the UV–Vis full-wavelength sensor reveals both the composition of the water sample and interference, such as high-frequency random noise, light scattering, and external light. Therefore, establishing a model with the original spectrum will likely jeopardize stability and accuracy due to irrelevant information and noise [23,24]. Hence, it is necessary to pre-process the original spectrum to eliminate any

interference [25]. In this study, the full-wavelength data were preprocessed using SG smoothing, the moving average method, first derivative, and second derivative preprocessing step by step. The effects of various preprocessing methods were evaluated based on the predictability of the model. The Unscrambler X 10.4 (64-bit) was used to preprocess the data. The correlation coefficient ($R^2$) between the prediction result and the actual value, and the root mean square of the prediction error (RMSEP) were used to evaluate the predictability of the model. The RMSEP was mainly used to evaluate the prediction error because it directly reflects the size of the prediction error. The smaller the RMSEP, the better the prediction effect and the stronger the predictability of the model. RMSEP was calculated using Equation (1):

$$\text{RMSEP} = \sqrt{\frac{\sum_{i=1}^{n_p} \left(y_i' - y_i\right)^2}{n_p}} \tag{1}$$

where $n_p$ is the number of validated samples and $y_i'$ and $y_i$ are the predicted and actual values of the $i$ sample in the test set, respectively.

The International Union of Pure and Applied Chemistry (IUPAC) recommends using Equations (2) and (3) to calculate the limits of detection [26]. In the formula, $x_L$ is the minimum analytical signal, $\overline{x_{b1}}$ is the mean value of the blank signal, $k$ is the constant related to confidence, $s_{b1}$ is the standard deviation of the blank signal, $c_L$ is the minimum detection concentration, and $S$ is the sensitivity of the method. The IUPAC recommends that the number of blank signals means $\overline{x_{b1}}$ measurements should not be less than 20 times. The value of $k$ is 3, and the confidence interval is 99.6.

$$x_L = \overline{x_{b1}} + ks_{b1} \tag{2}$$

$$c_L = \frac{x_L - \overline{x_{b1}}}{S} = \frac{ks_{b1}}{S} \tag{3}$$

### 2.3.2. Wavelength Selection

For methods that combine spectroscopy with multivariate correction, the conventional view is that multivariate correction methods (e.g., PLS) are more resistant to interference and can be modelled using the full range of wavelengths. With further research and application of methods such as PLS, it is possible to obtain better models by screening the characteristic wavelengths by specific methods. Wavelength selection simplifies the model and improves its operational efficiency on the one hand, and removes uncorrelated or non-linear variables on the other [27].

To select the optimal wavelength selection algorithm, the predictability of the model established using iPLS, SPA, and CARS was investigated. RMSEP, $R^2$, and the number of selected wavelengths were used as evaluation indicators. OpenSA for Python (available at https://github.com/FuSiry/OpenSA (accessed on 30 June 2022)) was used for wavelength selection using iPLS, SPA, and CARS. The iPLS method is a wavelength interval selection method that works by dividing the entire spectrum equally into a number of equal-width subintervals and performing the PLS regression on each subinterval to find the interval corresponding to the minimum cross-validation root-mean-square error (RMSECV) [28]. SPA is a forward cyclic selection method. It starts from one wavelength and counts the projection on the unselected wavelength in each cycle, eventually introducing the wavelength with the largest projection vector into the wavelength combination [29]. CARS is a Monte Carlo Sampling based method, which treats each variable as an individual and implements a stepwise elimination selection process for individual [30]. The above three methods have been widely used for the wavelength selection in UV–Vis spectroscopy. In most cases, CARS outperforms methods such as SPA and iPLS.

## 3. Results and Discussion

### 3.1. LODs of Antibiotics at Different Optical Path Lengths

The absorption spectra of the three antibiotics at different concentrations were scanned using Sensors 1 (Figure 1A–C) and 2 (Figure 1D–F). As shown in Figure 1, the absorption peaks of the three antibiotics selected in this experiment were all located in the near-ultraviolet region of 200–400 nm, and the 400–750 nm band was noise generated by the equipment. The UV–Vis absorption spectra are produced by the energy level transition of the valence electrons in the molecules. Therefore, the absorption spectrum depends on the nature of valence electrons. Tetracycline is a ketone- and enolol-conjugated double-bond system, and the carbonyl and conjugate systems produce R and K absorption bands. Therefore, tetracycline produces absorption peaks at 217–280 nm and 350–365 nm. The strongest absorption peak for ofloxacin was near 290 nm, with two shoulder peaks at 200–250 nm and 325–350 nm, respectively. The strongest absorption peak was located in the K absorption band, caused by the ketone and carbonyl groups. Chloramphenicol has an absorption peak near 280 nm and a terminal absorption at 200 nm. It contains p-nitrophenylethyl, hydroxymethyl, and hydroxy groups, and its main absorption peak is in the K absorption band. As chloramphenicol undergoes an n→σ* transition, it produces an absorption peak in the extreme ultraviolet band and only the terminal part is detected in the near-ultraviolet band. Both tetracycline and chloramphenicol exhibited characteristic absorption peaks at 280 nm. Tetracycline consists of two amino groups, four alcohol hydroxyl groups, three aldehyde groups, and one phenolic group. Chloramphenicol comprises an aldehyde group, two hydroxyl groups, and two chlorinated hydrocarbons. Both contained aldehyde groups and the absorbance of chloramphenicol was significantly higher than that of tetracycline at the same concentration. It is speculated that there is a relationship between the absorption peak of organic compounds at 280 nm and the existence of aldehyde groups, requiring further detailed studies. Ofloxacin has a more complex structure with one carbonyl group and a fluorinated hydrocarbon. Combined with the above analysis, the sensitivity of UV–Vis spectroscopy for the detection of organic compounds has no direct correlation with the reducibility, molecular weight, or chemical structural complexity of organic compounds. This is related to the extra-nuclear electron cloud formed by the active functional groups, and this phenomenon needs to be further explored. A special detection scene should be analyzed and modeled according to the spectral characteristics of the detection object.

In future, it will be possible to classify antibiotics by exploring the types of chromophores and, thus, to identify the types of antibiotics in hospital wastewater using UV–Vis spectroscopy. As shown in Figure 1, there was a slight difference in the position of the absorption peaks when measuring the same antibiotic on two different sensors. This is due to the different light sources used by the two sensors and the difference in the optical path length and temperature.

The linear fitting results for the absorbance and concentration of the three antibiotics are shown in Figure 2. There was a good linear correlation between the absorbance and the concentrations of tetracycline, ofloxacin, and chloramphenicol in the measured concentration range.

The slope of the fitting curve indicates the sensitivity of the detection method, which is the absorbance per unit mass concentration of antibiotics. In this experiment, the 2 sensors obtained 25 blank-signal values. The LODs of the two sensors for the three antibiotics were calculated according to Equation (3). Table 3 lists the calculation parameters and the results of the LODs at different characteristic wavelengths. When using Sensor 1, the LOD of ofloxacin at 290 nm and chloramphenicol at 280 nm were the lowest (0.4647 mg/L) and highest (1.1325 mg/L), respectively. When using Sensor 2, the LOD of tetracycline at 273 nm and chloramphenicol at 271 nm were the lowest (0.0031 mg/L) and highest (0.0954 mg/L), respectively. According to the Beer–Lambert law, when the optical path length increases, the absorbance of a substance also increases. As shown in Table 3, the LOD of Sensor 2 was significantly lower than that of Sensor 1. However, the optical path

of Sensor 2 was 10 cm, and the light intensity decayed rapidly when water was turbid. Therefore, it is necessary to choose the optimal optical path length according to actual water conditions.

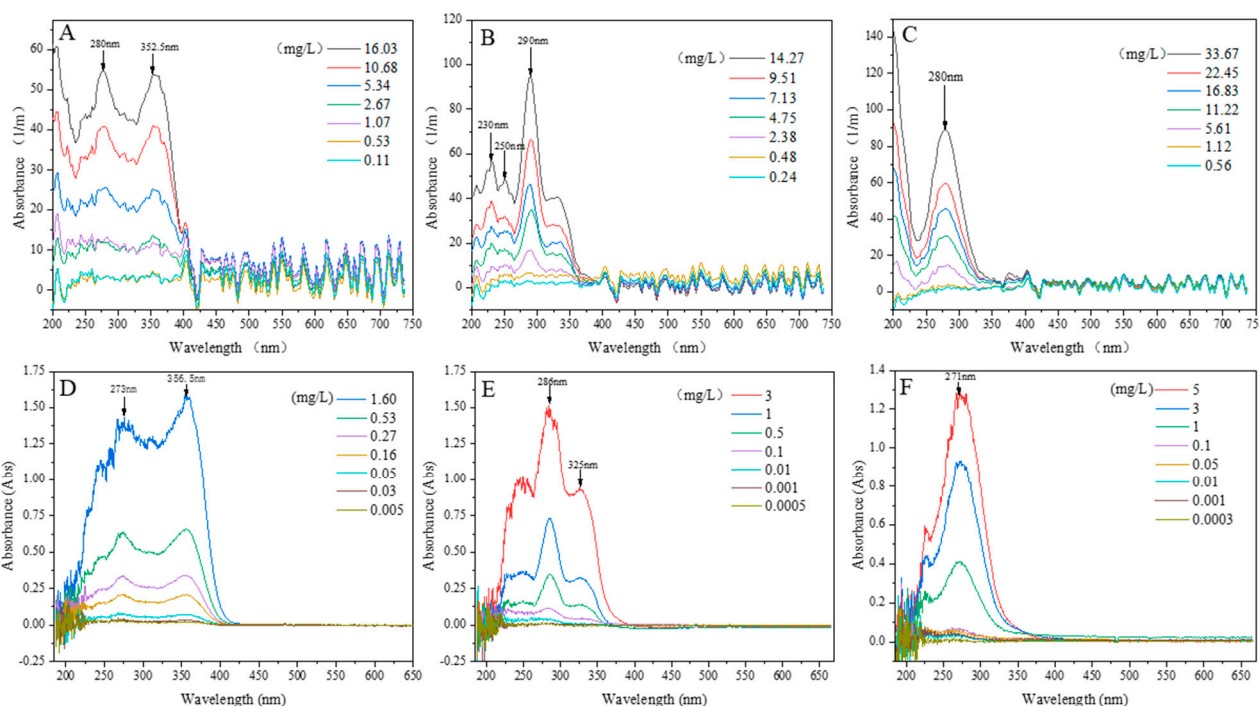

**Figure 1.** Absorption spectra of (**A**) tetracycline, (**B**) ofloxacin, and (**C**) chloramphenicol by Sensor 1 and (**D**) tetracycline, (**E**) ofloxacin, and (**F**) chloramphenicol by Sensor 2.

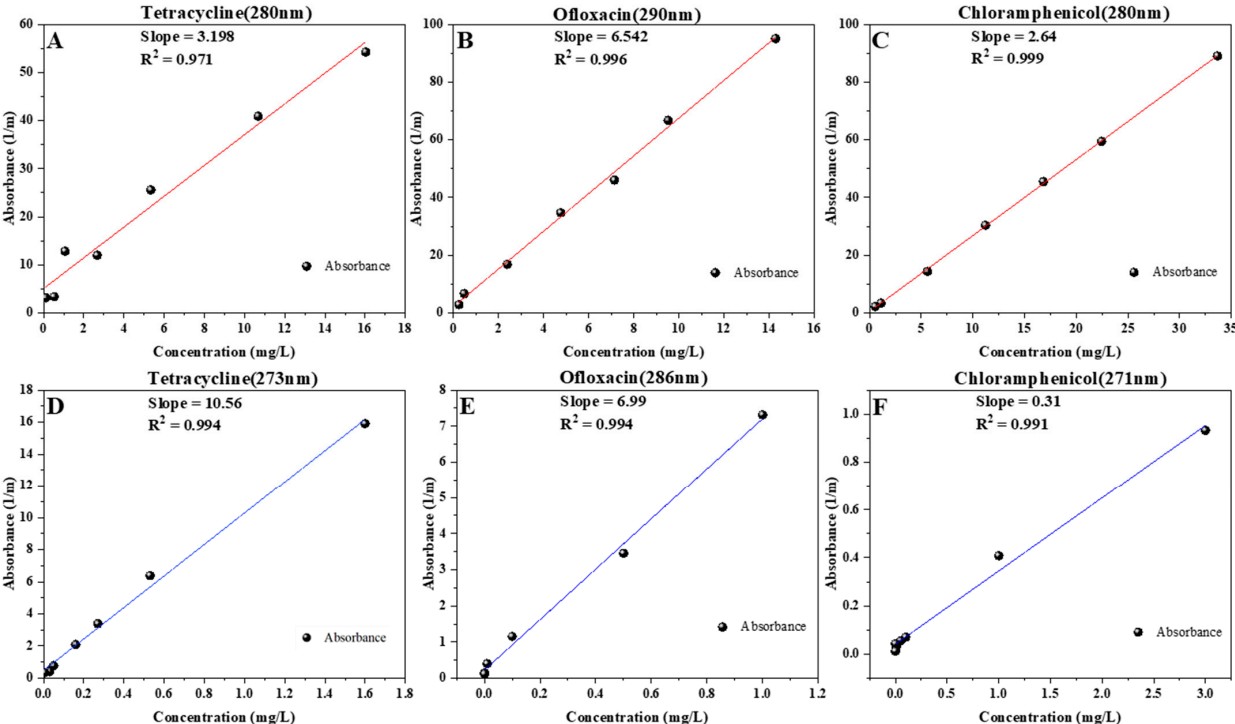

**Figure 2.** Linear fitting between the absorbance of three antibiotics by Sensor 1 (**A**–**C**) and 2 (**D**–**F**) and concentration at the absorption peak.

**Table 3.** Calculation parameters and limits of detection at different characteristic wavelengths.

| | Mean Value of Blank Sample (1/m) | Standard Deviation of Blank Sample (1/m) | Sensitivity | Limit of Detection (mg/L) |
|---|---|---|---|---|
| 280 nm (Tetracycline) [1] | 1.26604 | 0.99657 | 3.198 | 0.9349 |
| 290 nm (Ofloxacin) [1] | 1.07527 | 1.01339 | 6.542 | 0.4647 |
| 280 nm (Chloramphenicol) [1] | 1.26604 | 0.99657 | 2.640 | 1.1325 |
| 273 nm (Tetracycline) [2] | −0.00429 | 0.01089 | 10.560 | 0.0031 |
| 286 nm (Ofloxacin) [2] | −0.00259 | 0.01052 | 6.990 | 0.0045 |
| 271 nm (Chloramphenicol) [2] | −0.00159 | 0.00986 | 0.310 | 0.0954 |

[1] Measure by Sensor 1. [2] Measure by Sensor 2.

### 3.2. Detection of Antibiotics in Wastewater Using UV–Vis Spectroscopy

All optical detection methods are affected by suspended solids in hospital wastewater, causing light to be refracted or scattered. Suspended solid concentrations in hospital wastewater are within the range of 40–120 mg/L according to the Technical Guidelines for Hospital Wastewater Treatment of China [21]. In this study, the water quality of hospital wastewater was simulated using wastewater treatment plant influent. However, turbidity affects the light penetration of the sensor; therefore, tests were conducted to determine the appropriate optical path length. Firstly, the LODs of antibiotics in distilled water at 10, 5, 3, and 1 cm optical path lengths were calculated. The results are shown in Table 4. The optical path length shows a negative correlation with the LOD, but there is no linear relationship. As the length of the light path gets longer, the weakening of the light as it passes through the liquid becomes more pronounced. The presence of SS in the wastewater causes a more pronounced attenuation of light, so the suitable light path length needs to be selected.

**Table 4.** LODs of three antibiotics in distilled water at different optical path lengths.

| Optical Path Length (cm) | Limit of Detection (µg/L) | | |
|---|---|---|---|
| | Tetracycline | Ofloxacin | Chloramphenicol |
| 10 | 3.1 | 4.5 | 95.4 |
| 5 | 18.4 | 25.3 | 137.5 |
| 3 | 43.7 | 58.9 | 173.8 |
| 1 | 75.4 | 95.1 | 241.2 |
| 0.05 | 934.9 | 464.7 | 1132.5 |

The incident luminous intensity is the light intensity detected after the xenon lamp light passes through the reference optical path, and the transmission luminous intensity is the light intensity after the xenon lamp light passes through the liquid to be measured. In Figure 3, the incident and transmission luminous intensities were tested at lengths of 10, 5, 3, and 1 cm. As shown in Figure 3A, when the optical path length was 10 cm, the transmission luminous intensity was weakened completely, and the light intensity of the xenon lamp light source used in Sensor 2 was weaker in the near-ultraviolet band than that in the visible band. When the optical path length was 10 cm, light was completely absorbed in the 200–350 nm range, i.e., the upper measurement limit was reached in that range. As shown in Figure 3B,C, this range becomes 200–300 nm and 200–250 nm when the optical path lengths are 5 and 3 cm, respectively. This phenomenon shows that as the length of the optical path decreases, the refraction and scattering of light by SS in the wastewater gradually decreases. When choosing the 1 cm optical path length, the entire near-ultraviolet band did not reach the upper detection limit. Therefore, an optical path length of 1 cm was used to measure the actual wastewater.

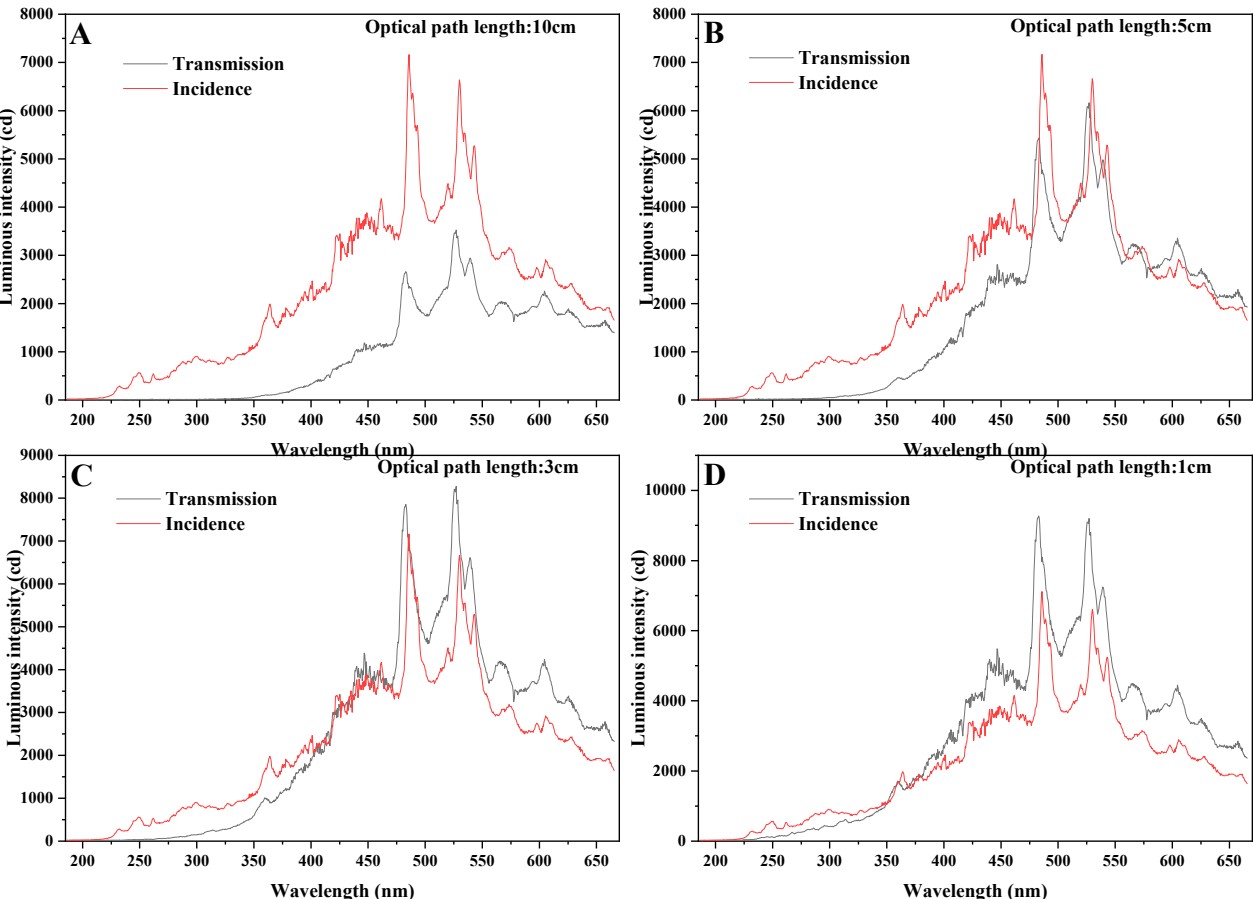

**Figure 3.** Comparison of incident and transmission luminous intensities for (**A**) 10 cm, (**B**) 5cm, (**C**) 3 cm, and (**D**) 1 cm optical path lengths.

Figure 4 shows a decreasing curve from the short-wavelength to the long-wavelength direction for the wastewater without added antibiotics. This implies that the LODs for antibiotics with absorption peaks in the near-ultraviolet band are limited. In wastewater, the absorption peak of tetracycline is located at 372 nm, whereas in distilled water, it is located at 356.5 nm. The positions of the absorption peaks of ofloxacin and chloramphenicol remained essentially unchanged. Only tetracycline showed a bathochromic shift in this study, which could be due to a change in the polarity of the solution or a change in the spatial structure of the molecule. The method in Section 3.1 calculates the LOD for antibiotics in actual wastewater. The LOD for tetracycline, ofloxacin, and chloramphenicol were 0.094, 0.107, and 0.264 mg/L, respectively. Even in untreated wastewater, this method still had a significant LOD. However, compared to the LOD of antibiotics in distilled water at 1 cm optical path in Table 4, the LOD in wastewater is higher. The reason for this is that the high concentration of SS in the wastewater causes severe light weakening. Hospital wastewater does not reach such high SS concentrations; therefore, the LODs will be lower when applied in hospital wastewater.

Relevant studies have established that the use of antibiotics in China remains high despite the promotion of the "smart use of antibiotics" policy [31]. Li et al. [32] analyzed the medical records of 59 general hospitals in China between 2012 and 2016, of which 70.27% of inpatient medical records used antibiotics. The complex and high concentrations of antibiotics in hospital wastewater may make it possible to monitor the concentration of a particular class of antibiotics rather than a particular one by UV–Vis spectroscopy. Both pharmaceutical and hospital wastewater contain high concentrations of antibiotics. Tang et al. [33] tested the concentration of antibiotics in a pharmaceutical wastewater treatment plant and showed that the concentration of cephalosporin antibiotics in the influent water

of the plant was up to 119 μg/L. This means that the method proposed in this study can be applied to the online monitoring of hospital and pharmaceutical wastewater. Further research should be directed towards developing light sources with excellent transmittance in wastewater.

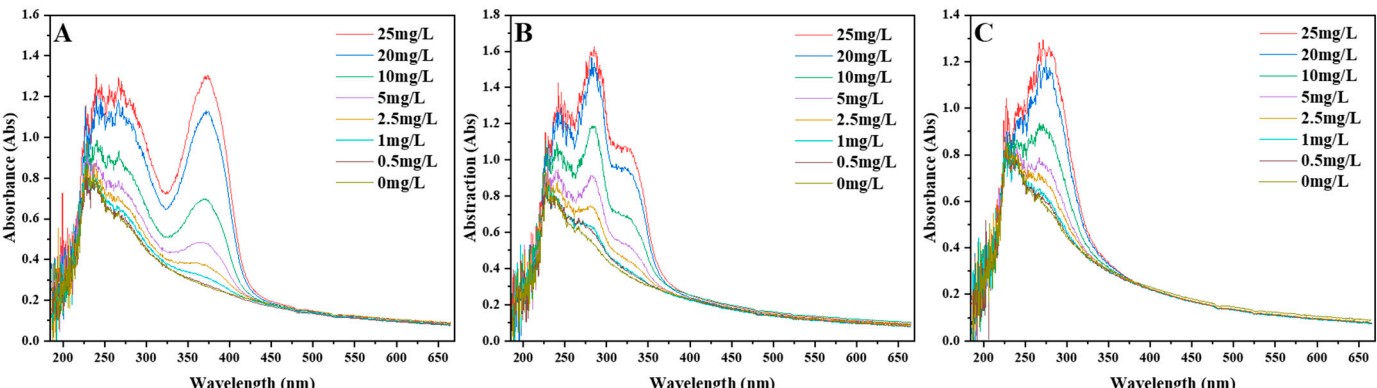

**Figure 4.** Absorption spectra of (**A**) tetracycline, (**B**) ofloxacin, and (**C**) chloramphenicol diluted to different concentrations with wastewater.

### 3.3. Model Establishment of Mixed Samples

#### 3.3.1. Data Preprocessing

The predictability of the model without data preprocessing and with SG smoothing, the moving average method, and first- and second-derivative preprocessing were investigated. The RMSEP, $R^2$, and number of latent variable (LVs) for the three components analyzed using different pretreatment methods are listed in Table 5. As shown in Table 5, different components are suitable for using different pretreatment methods when modeling, and the evaluation is based on the premise that the RMSEP is as small as possible, $R^2$ is closer to 1, and the number of hidden variables is small. Among them, the best results were achieved with the second derivative pretreatment with ofloxacin, the first derivative pretreatment with tetracycline, and the moving average pretreatment with chloramphenicol.

**Table 5.** LVs, RMSEP, and $R^2$ using different preprocessing methods.

| Pretreatment Method | Ofloxacin | | | Tetracycline | | | Chloramphenicol | | |
|---|---|---|---|---|---|---|---|---|---|
| | LVs | RMSEP | $R^2$ | LVs | RMSEP | $R^2$ | LVs | RMSEP | $R^2$ |
| No pretreatment | 4 | 3.16105 | 0.98799 | 3 | 1.51881 | 0.99699 | 3 | 3.33167 | 0.99015 |
| SG smoothing | 4 | 3.46094 | 0.98571 | 3 | 1.55338 | 0.99594 | 5 | 3.3529 | 0.99011 |
| Moving average | 3 | 3.02822 | 0.98959 | 4 | 1.56447 | 0.99673 | 3 | 0.80787 | 0.99941 |
| First derivative | 4 | 3.28151 | 0.98764 | 3 | 0.84661 | 0.99905 | 5 | 0.72231 | 0.99957 |
| Second derivative | 4 | 2.28809 | 0.99311 | 5 | 1.84422 | 0.99519 | 5 | 4.46114 | 0.98245 |

#### 3.3.2. Wavelength Selection

Based on the above pretreatment methods, which are suitable for different antibiotics, different wavelength selection algorithms were compared (Table 6). Due to the large randomness of CARS, it was run five times to record the wavelength selected at least thrice for PLS modeling. It can be observed from Table 6 that the SPA method can select a small number of wavelengths. This is due to the use of vector projection analysis, projecting wavelengths onto other wavelengths, and comparing the size of the projection vector. The wavelength with the largest projection vector is selected as the wavelength, and the final characteristic wavelength is selected based on the correction model. The SPA chooses a combination of variables with the least redundant information and the least collinearity. The model established after the above three wavelength selection algorithms for chloramphenicol was not ideal; the RMSEP was high, and the correlation coefficient

was generally lower than 0.99. The reason for this result is that chloramphenicol only has a unique absorption peak in the wavelength range of 200–400 nm, whereas the characteristic absorption peaks of ofloxacin and tetracycline are not unique, leading to a poor prediction of chloramphenicol.

**Table 6.** Ws, RMSEP, and $R^2$ using different wavelength selection methods.

| Wavelength Selection Method | Ofloxacin | | | Tetracycline | | | Chloramphenicol | | |
|---|---|---|---|---|---|---|---|---|---|
| | Ws [1] | RMSEP | $R^2$ | Ws [1] | RMSEP | $R^2$ | Ws [1] | RMSEP | $R^2$ |
| SPA | 5 | 3.30572 | 0.98800 | 5 | 2.45338 | 0.99094 | 3 | 8.51275 | 0.93337 |
| CARS | 16 | 2.49126 | 0.99281 | 6 | 0.84635 | 0.99873 | 13 | 3.74908 | 0.98767 |
| iPLS | 9 | 1.88800 | 0.99528 | 9 | 1.75843 | 0.99600 | 28 | 4.13771 | 0.98305 |

[1] Number of wavelengths selected.

The SPA algorithm for ofloxacin, which was the lowest among the three algorithms, selected five wavelengths; however, the RMSEP was relatively high. The nine wavelengths selected by iPLS were within the allowable range, RMSEP was the lowest, and $R^2$ was the highest. Therefore, iPLS was chosen as the wavelength selection method for ofloxacin. Similarly, for tetracycline and chloramphenicol, CARS was selected for wavelength selection and modeling, so that the number of variables used to establish the model was greatly reduced. For ofloxacin and tetracycline, the modeling effect after the wavelength selection was significantly improved.

Insausti et al. [34] acquired UV–Vis, near-infrared (NIR), and synchrotron fluorescence (SF) spectra of 70 samples composed of 3 different substances and differentiated the substances by soft independent modeling of class analogies and linear discriminant analysis with variables selected by the successive projection algorithm (SPA-LDA). The SPA-LDA can efficiently select wavelengths and differentiate between different substances. However, compared with the other two spectra, the UV–Vis spectrum was selected with more wavelengths, contrary to the results of this experiment. The reason for this phenomenon may be that the UV–Vis spectrum in this experiment had obvious characteristic absorption peaks. Xu et al. [35] used CARS and PLSR methods to determine mixed water samples containing divalent metal ions, cadmium, zinc, and cobalt. This method can reduce the 916 wavelengths to a maximum of 21. The experimental results show that the wavelengths selected by the CARS are mostly located at the extreme point of the first derivative. Yu et al. [36] used all the data in the range of 230–400 nm to build the model by combining dynamic orthogonal projection correction and a support vector machine and introduced a Bayesian anomaly detection method. The semi-supervised learning model achieved good results; however, the process was complicated and unsuitable for online monitoring. Guo et al. [37] first increased the concentration of trace organics by solid-phase extraction and then performed PLS modeling on 401 absorbance data in the range of 200–600 nm. This method can effectively detect water samples containing fewer organic compounds, with no obvious overlapping absorption peaks.

Figure 5 shows the characteristic variables obtained from the wavelength selection of ofloxacin, tetracycline, and chloramphenicol using iPLS, CARS, and CARS. Nine characteristic variables were selected by iPLS, and 6 and 13 characteristic variables were selected by CARS for tetracycline and chloramphenicol, respectively. As shown in Figure 5, because the characteristics of iPLS only choose a continuous interval and the selected interval coincides with the strongest characteristic absorption peak of ofloxacin, the absorption spectrum of the refractory organic mixed solution remains affected by substances with high absorbance. Conversely, the selected wavelengths of tetracycline were located in the position of rapid change in absorbance, which is related to the first-derivative pretreatment of the tetracycline original data. The selected wavelength of chloramphenicol was dispersed in the range of 200–350 nm, whereas there was a concentrated characteristic wavelength in the wavelength range of 280–330 nm; this corresponds to the position of its characteristic

absorption peak. Therefore, the mixed solution retained some of the characteristics of the individual components. This analysis demonstrated the importance of wavelength selection in complex systems by reducing interference and improving predictability.

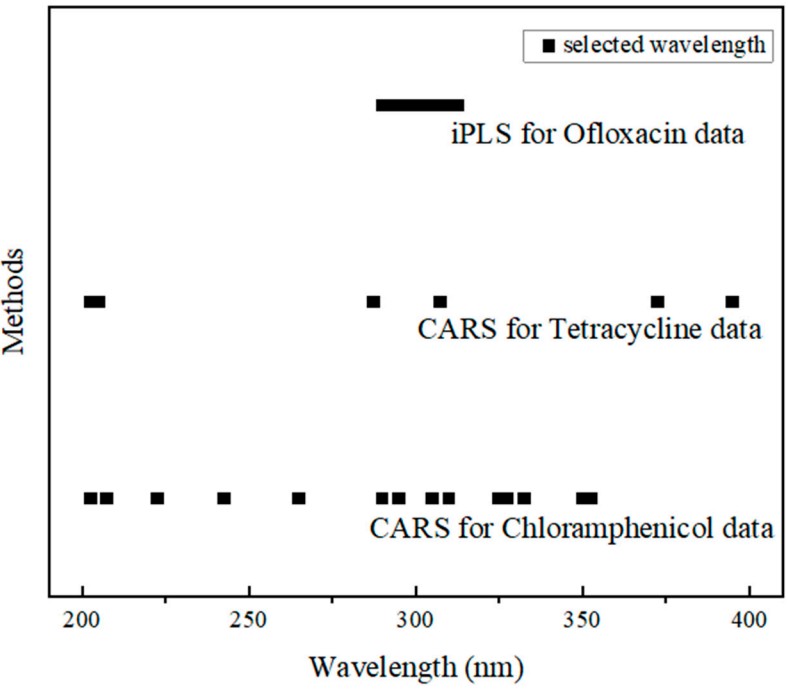

**Figure 5.** Wavelength selected with iPLS, CARS, and CARS for the three components.

In this study, PLS was used for modeling because it overcomes the weaknesses of multiple linear regression in terms of its lack of rank inversion and inadequate use of spectral information. PLS can decompose both the spectral and concentration matrices and consider the interrelationships between them to ensure the best model [38]. Tetracycline, ofloxacin, and chloramphenicol discussed in this experiment are typical antibiotics that are frequently detected in surface water and cause serious harm to the ecological environment. Therefore, it is necessary to control the source of antibiotics, such as hospitals, pharmaceutical factories, etc. Common methods for the treatment of antibiotics include hydrogen peroxide oxidation, ozone oxidation, and electrochemical technology; however, there is no method to ensure the stable degradation of all antibiotics [39]. Therefore, the development of a method for online monitoring of the concentration of antibiotics described in this paper can help an advanced treatment system to adjust ozone aeration or drug dosage in time according to the monitoring results to achieve automatic control. Currently, technologies for detecting the concentration of antibiotics using UV–Vis spectroscopy have not been developed in the industry. According to the above, the LOD of antibiotics determined by UV–Vis spectroscopy is as low as the microgram level. For example, the LOD of tetracycline can reach 3.1 µg/L, which is significantly lower than the antibiotic concentration in hospital and pharmaceutical wastewater. Even though actual wastewater usually contains more organic matter and is disturbed by factors such as turbidity and chromaticity, this method still has a low LOD but cannot be detected according to the characteristic wavelengths of some antibiotics. Therefore, an experiment was designed to detect three types of synthetic wastewater samples containing antibiotics with overlapping absorption peaks. The original data were pretreated and wavelength-selected, and models were established. The purpose of this study was to explore the ability of UV–Vis spectroscopy to predict the components of antibiotics in synthetic wastewater samples. A cross-validation method was used to evaluate the predictability of the model, and it was shown that the model had good predictability for each component in the synthetic wastewater sample. UV–Vis spectroscopy has obvious advantages over traditional detection methods for antibiotics,

such as gas chromatography-mass spectrometry (GC-MS), because it can both monitor in situ and measure data quickly.

## 4. Conclusions

We propose a novel method that combines a preprocessing algorithm, wavelength selection algorithm, and PLS to measure the concentration of each component in mixed water samples composed of ofloxacin, tetracycline, and chloramphenicol. The LODs of the three antibiotics were calculated by configuring a standard solution to determine the absorbance of the characteristic absorption peaks. The results showed that the average LOD for the three antibiotics was 0.85 mg/L when using sensor 1 with a 0.5 mm optical path length, and the lowest LOD was 0.4647 mg/L based on the absorption peak of ofloxacin at 290 nm. The lowest LOD was 3.1 μg/L for antibiotics when using Sensor 2 with a 10 cm optical path length. An LOD of 94 μg/L was also achieved in artificially simulated wastewater. These results can meet the monitoring needs of specialized production wastewater facilities, such as medical and pharmaceutical wastewater. The absorption spectra of tetracycline, ofloxacin, and chloramphenicol synthetic wastewater samples with overlapping peaks were analyzed, and the pretreatment and wavelength selection methods most suitable for each component were selected for modeling. The results show that the RMSEP of tetracycline, ofloxacin, and chloramphenicol using this method were 0.84635, 1.888, and 3.7491, respectively, and $R^2$ was greater than 0.98.

This study demonstrates that UV–Vis spectroscopy has an LOD level that meets engineering applications and can simultaneously monitor multiple antibiotics in combination with chemometrics. However, SS and chromaticity in the wastewater will prohibit light transmission, so the LOD of spectroscopy will not narrow down by increasing the optical path length solely. Moreover, straight-chain alcohols do not exhibit characteristic absorption in the UV–Vis region, also limiting their application. In conclusion, UV–Vis spectroscopy is well suited to the monitoring of antibiotics in hospital and pharmaceutical wastewater, but its application in other water environments is yet to be developed.

**Author Contributions:** Conceptualization, X.W.; Data curation, F.L.; Formal analysis, F.L.; Funding acquisition, X.W. and H.R.; Investigation, F.L. and M.Z.; Methodology, F.L., X.W. and M.Y.; Project administration, Q.L.; Resources, D.S.; Software, M.Z.; Supervision, X.B. and H.R.; Validation, X.W.; Visualization, W.C.; Writing—original draft, F.L.; Writing—review and editing, X.W., M.Y. and Z.M. All authors have read and agreed to the published version of the manuscript.

**Funding:** This research was funded by The National Natural Science Foundation of China (grant number: 51908303) and The Research Council of Norway (grant number: 310074/G10).

**Institutional Review Board Statement:** Not applicable.

**Informed Consent Statement:** Not applicable.

**Data Availability Statement:** Not applicable.

**Acknowledgments:** This project was supported by the Urban Water Pollution Control and Recycling Research Group of Qingdao University of Technology.

**Conflicts of Interest:** The authors declare no conflict of interest.

## Appendix A

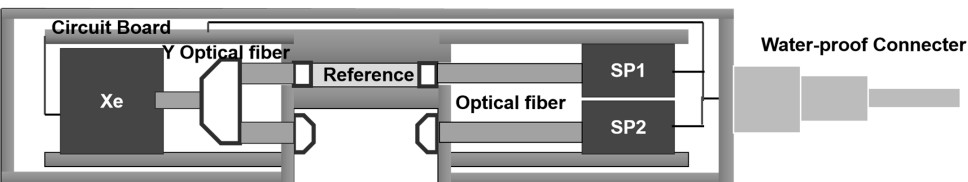

**Figure A1.** The scheme of Sensor 2.

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
