# Peer review of "Detection Limits of Antibiotics in Wastewater by Real-Time UV–VIS Spectrometry at Different Optical Path Length"

_processes, doi:10.3390/pr10122614_

Round 1
Reviewer 1 Report
The proposed paper is devoted to a subject of interest nowadays: the detection of antibiotics , more general, medicines in waste water at as low as possible concentrations and in real-time, using relatively simple and cheap devices.
As is presented here, the paper may be accepted for publication after major revision, considering the following, but not exclusive, observations:
1.The authors use in their measurements specific sensors, out of which, one is developed by themselves. It is recommended that they show a scheme of the used devices.
2.At subchapter “2.2. Wastewater samples”, the authors mention that measurements were made using two sensors out of which sensor 2 is developed by themselves but they do not show results obtained with this sensor. More, they mention “However, because the two sensors have different upper absorbance limits, the concentration ranges of the measured samples are also different.” Then, they present only the concentration range covered with sensor 1. This information looks incomplete and it is not convincing. Please reformulate the text and the data presentation.
3.In subchapter ” 3.1. LODs of antibiotics at different optical path lengths” the authors mention “As shown in Figure 1, there was a slight difference in the position of the 259 absorption peaks when measuring the same antibiotic on two different sensors. This is 260 because of the different light sources used by the two sensors and the difference in the 261 optical path length and temperature.” In order to provide more accurate information, it is recommended that they show the emission spectra of the two light sources in the two sensors, respectively, to observe the differences that might justify these conclusions.
4.In subchapter “3.2. Detection of antibiotics in wastewater using UV–Vis spectroscopy “, it is mentioned “ In Figure 3, the incident and transmission luminous intensities were tested at lengths 313 of 10, 5, 3, and 1 cm.” The authors should define what do they mean by incident and transmission intensities, considering the spectra shown in Figure 3 and the comparison of A and B spectra with C and D.
5. English typing and grammar errors should be corrected.
Reviewer 2 Report
In their manuscript, Li et al. present a method based on UV-Vis spectroscopy for the quantitative detection of antibiotics in wastewater. They use two immersed sensors with different characteristics to analyze synthetic wastewater (a mixed water sample containing three antibiotics) and actual wastewater. The best pretreatment and wavelength selection methods are identified and the limits of detection for each antibiotic are established. It is believed that such a method could allow faster in situ testing of medical and pharmaceutical waters, at lower costs.
The paper is well written and scientifically sound, and I consider that it can be published in Processes in its present form. Some minor observations that should be addressed by the authors are listed below.
It should be mentioned in the caption of Table 2 or as a table footnote which data are taken from ref. 21 and 22.
pg. 5, line 201: “For methods that spectroscopy combined with multivariate correction” should say “For methods that combine spectroscopy with multivariate correction”
pg. 14, line 491: “the spectroscopy method cannot to extend the longer optical path length” - please rewrite for clarity
pg. 10, line 327: When discussing the absorption features of tetracycline, the authors state that its spectrum in wastewater is shifted with respect to its spectrum in reference distilled water by about 16 nm (372 vs. 356.5 nm). I am concerned on how such difficult to predict shifts that occur (due to polarity, pH, conformational changes, etc.) in real systems containing a large number of constituents can affect the accuracy of identification and quantitative detection. Do the authors have an estimate as to which extent such effects might alter the data, and also regarding which of the pretreatment methods discussed here could be more useful in providing data less prone to errors?
Reviewer 3 Report
In this work, a study on the detection limits of antibiotics in wastewater by real-time UV-VIS spectrometry at different optical path length was developed.
This manuscript is an interesting work on the develop of a novel method that combines a preprocessing algorithm, wavelength selection algorithm, and PLS to measure the concentration of each component in mixed water samples composed of ofloxacin, tetracycline, and chloramphenicol. This study demonstrates that UV–Vis spectroscopy has an LOD level that meets engineering applications and can simultaneously monitor multiple antibiotics in combination with chemometrics. Besides, this manuscript is on the scope of the journal and is of board interest. However, the information presented requires major minor in view of improving some aspects. General comments have been provided below.
Page 4, line 146-148: This paragraph is unclear and would benefit from rephrasing.
Page 4, line 139-140. The authors indicate that "the antibiotics were diluted in the influent wastewater from the wastewater treatment plants (EDAR) to simulate the wastewater from a hospital", however, it is not very clear, in the manuscript, which is the dilution procedure and their concentrations. There should be more clarity on this aspect.
Pag 5, line 190-197. The bibliographical reference on the IUPAC recommendations regarding the calculation of the detection limit should be cited.
Round 2
Reviewer 1 Report
The paper may be accepted as is.